# Enhanced Neural Architecture for Real-Time Deep Learning Wavefront Sensing

**DOI:** 10.3390/s25020480

**Published:** 2025-01-16

**Authors:** Jianyi Li, Qingfeng Liu, Liying Tan, Jing Ma, Nanxing Chen

**Affiliations:** 1Free-Space Optical Communication Technology Research Center, Harbin Institute of Technology, Harbin 150001, China; 22b921018@stu.hit.edu.cn (J.L.); tanly@hit.edu.cn (L.T.); majing@hit.edu.cn (J.M.); 24b321003@stu.hit.edu.cn (N.C.); 2National Key Laboratory of Laser Spatial Information, Harbin Institute of Technology, Harbin 150001, China

**Keywords:** real-time wavefront sensing, deep learning, CNN, multi-objective neural architecture search, atmospheric turbulence

## Abstract

To achieve real-time deep learning wavefront sensing (DLWFS) of dynamic random wavefront distortions induced by atmospheric turbulence, this study proposes an enhanced wavefront sensing neural network (WFSNet) based on convolutional neural networks (CNN). We introduce a novel multi-objective neural architecture search (MNAS) method designed to attain Pareto optimality in terms of error and floating-point operations (FLOPs) for the WFSNet. Utilizing EfficientNet-B0 prototypes, we propose a WFSNet with enhanced neural architecture which significantly reduces computational costs by 80% while improving wavefront sensing accuracy by 22%. Indoor experiments substantiate this effectiveness. This study offers a novel approach to real-time DLWFS and proposes a potential solution for high-speed, cost-effective wavefront sensing in the adaptive optical systems of satellite-to-ground laser communication (SGLC) terminals.

## 1. Introduction

Over recent decades, satellite laser communication (SLC) has emerged as a focal point of research due to its numerous advantages, including high data rates, extended communication distances, improved confidentiality, excellent beam directivity, and a license-free spectrum [1,2]. Among the various SLC systems, the satellite-to-ground laser communication (SGLC) link is particularly significant. However, the communication channel is significantly degraded by atmospheric turbulence [3,4], for which adaptive optics (AO) serves as an effective solution for wavefront compensation of distortions [5]. Wavefront sensors, such as Shack–Hartmann wavefront sensors, are integral components of AO systems. However, these sensors also pose challenges due to their large size, high cost, substantial measurement losses at sub-aperture levels, and poor performance under conditions of strong scintillation [6,7].

The rapid advancement of artificial intelligence and the enhancement in computational unit performance have sparked a surge in interest in deep learning wavefront sensing (DLWFS). This method has become prominent due to its simplicity, speed, and high precision [8,9]. Currently, the primary implementation strategies can be divided into two categories: The first leverages deep neural networks (DNN) for iterative optimization of wavefront compensation, adhering to the physical model of beam propagation [10,11]. However, this method is hindered by time-consuming iterations during application. The second strategy employs DNNs to directly learn the mapping function between light signal intensity and wavefront from data, thereby circumventing the time-intensive iterative optimization process during application [12,13]. In the second implementation scheme, many current works focus on classic image classification convolutional neural network (CNN) prototypes, which modify the model for fitting Zernike mode coefficients of the wavefront [14,15,16]. While these approaches have proven effective, the parameter amounts of CNN prototypes are enormous (in the millions), resulting in longer inference times (5–20 ms). This consumes substantial hardware resources and power, increases servo latency, and poses challenges in adapting to real-time DLWFS in complex and dynamic environments of atmospheric turbulence.

This paper addresses the real-time sensing problem of dynamic random wavefront distortions caused by atmospheric turbulence by proposing a real-time description method based on CNNs. We aim to achieve low-cost, miniaturized, and fast real-time DLWFS through improvements to the neural architecture of the CNN-based wavefront sensing neural network (WFSNet). By improving and applying the multi-objective neural architecture search (MANS) methodology [17], we identified significant potential for improvement in classic CNN prototypes for wavefront sensing applications. A real-time DLWFS scheme for SGLC communication systems is proposed, establishing a DLWFS physical model and presenting a straightforward and effective automated design and optimization scheme for WFSNet. A statistical model for wavefront distortion caused by sensor noise and atmospheric turbulence is developed, providing a theoretical foundation for the effectiveness of real-time DLWFS simulations. Through training and structural optimization based on the classic EfficientNet-B0 prototype [18], we introduce an optimized WFSNet model: Efficient-WFSNet. Simulation results indicate that for the first 35 Zernike modes of distortion, the accuracy of wavefront sensing improved by 22%, while FLOPs and inference time were reduced by 80% and 95%, respectively. The robustness of the method was studied and validated under varying intensities of atmospheric turbulence and sensor noise. Further validation through indoor experiments demonstrates that this model demonstrates superior performance. To our knowledge, the optimized WFSNet achieves an inference time significantly below 1 ms (approximately 0.2 ms), representing an improvement of at least one order of magnitude compared to the levels of other current works [10,11,12,13,14,15,16]. This work provides a novel approach for real-time DLWFS and offers a new method for developing low-loss, miniaturized, and low-cost adaptive wavefront compensation in SGLC terminals, with advantages such as reduced computational load, lower hardware dependency, and cost-effectiveness, thus holding significant application value.

## 2. Principle of the Method

### 2.1. Configuration

A typical SGLC terminal receiver consists of two main subsystems: the pointing, acquisition, and tracking (PAT) subsystem and the communication subsystem. The primary function of the PAT subsystem is to establish and maintain a stable optical link. Subsequently, the communication subsystem receives the incident light beam affected by turbulence and converts it into electrical signals. Figure 1 illustrates the wavefront sensing scheme used for AO correction within the communication subsystem. The incident light beam is split into two beams by a beam splitter (BS). One beam is directed for wavefront sensing and passes through a receiving lens into a camera. The CNN model processes the spot input and outputs wavefront distortion measurement signals, which are then converted into voltage control signals. The piezoelectric ceramic elements of the DM receive voltage control signals from the driver to correct the wavefront on the pupil plane. The other beam, intended for communication, is coupled into an optical fiber for demodulation.

Our approach employs a single-shot measurement at the defocused plane. Research has shown that DLWFS exhibits superior performance on scattered, overexposed, and defocused spots compared to focused spots [13]. In the context of SGLC systems, implementing defocusing is straightforward, causes little energy loss, and does not impact the speed of sensing.

### 2.2. Problem Modeling

The ideal light field emitted by the transmitter propagates through the atmospheric channel, resulting in wavefront distortion at the receiver plane. The complex amplitude of incident beam at the receiver plane can be expressed as U(x,y)=A(x,y)exp[jW(x,y)], where A(x,y) is the aperture function, equal to 1 in a certain region and 0 elsewhere. W(x,y) represents the wavefront distortion, which can be expressed as a weighted sum of a series of Zernike polynomials $Z_{i}\left(x,y\right)$ in the case of a circular aperture, i.e., $W\left(r\right)=\sum_{i=1}^{\infty }a_{i}Z_{i}\left(x,y\right)$. Under the Fresnel approximation, the intensity pattern $I(x^{\prime },y^{\prime })$ captured by the camera after propagating through a lens (focal length f) with defocus $f_{d}$ is expressed as follows: (1)$$\begin{array}{cc} I(x^{\prime },y^{\prime }) & =A{\left|{\;\int \int \;}_{-\infty }^{\infty }U\left(x,y\right)\cdot \exp\left[j\beta \left(x^{2}+y^{2}\right)\right]\cdot \exp\left[-j2\pi \left(xu+yv\right)\right]\;dxdy\right|}^{2}\\ \; & =A{\left|F\left\{U\left(x,y\right)\cdot \exp\left[j\beta (x^{2}+y^{2})\right]\right\}\right|}^{2} \end{array}$$
where $\beta =\pi (1/f+1/f_{d})/\lambda$ is a factor related to defocus, which is equivalent to Zernike coefficient $a_{4}$. *A* is a real constant, $u,v=x^{\prime }/\lambda \left(f+f_{d}\right),y^{\prime }/\lambda \left(f+f_{d}\right)$ are spatial frequencies, and λ is the wavelength. The principle of the DLWFS scheme can be described as follows: to find a target mapping *h* such that the intensity distribution *I* is mapped to the *N*-dimensional Zernike coefficient vector $a_{N}$ of the wavefront distortion. The error is denoted as E, and the optimization process can be formulated as follows:(2)$$\min_{h}E\left[h\left(I\right)-a_{N}\right]$$

In this scheme, a CNN is employed to represent the target mapping *h*, and the model parameters are optimized using the backpropagation algorithm.

Noll [19] has demonstrated, based on the Kolmogorov model, that atmospheric-induced wavefront distortions are concentrated in the lower-order Zernike modes. Research indicates that for single-mode fiber coupling, when the corrected Zernike polynomial terms exceed 35, the improvement in coupling efficiency becomes negligible [6]. The correction requirement can be further relaxed for multimode fibers with larger core diameters [20]. Therefore, in our study, we set N=35. However, in practical environments, the Zernike mode fitting of distorted wavefronts will include an infinite number of terms. Therefore, in the analysis of phase estimation interference, it is essential to consider the random distortions caused by Zernike modes greater than *N*.

### 2.3. Model Structure Design and Optimization Method

For the task of real-time sensing of dynamic random wavefront distortions caused by atmospheric turbulence, the architecture of the CNN model *h* is critical to its performance. Key factors include the number of convolutional layers, the kernel size, the number of filters, stride, activation functions, and other structural hyperparameters. These factors are interrelated and collectively determine the performance of the CNN model. Our optimization objective is to explore the structural hyperparameters to achieve an optimal balance between computational complexity and generalization performance, referred to as multi-objective neural architecture search (MNAS) [17]. Traditionally, three main implementation methods exist [21]: grid search, random search, and reinforcement learning approaches. However, these methods require substantial computational resources and time, making them difficult to adapt to different optical systems and noise environments. Therefore, we propose a streamlined yet effective method that leverages the inherent characteristics of CNNs to reduce the number of uncertain hyperparameters, while simultaneously utilizing local gradient information during the search process.

Research by Tan et al. [18] indicates that the connection between neurons in CNN is efficient when the resolution (size of input images and hidden layers), depth (number of hidden layers), and width (number of channels in each hidden layer) are scaled proportionally. Therefore, once the size of the input image is determined, the structural hyperparameters to be specified can be simplified to the depth d and the width scaling factor w. Additionally, when other important components are present within the CNN, corresponding structural hyperparameters can be incorporated. Once the hyperparameters are established, a grid search space will be defined, with each point Θ representing a structural choice, and the intervals δ determined by the user’s practical requirements. Typically, these hyperparameters are integers, making a discretized search space more meaningful. Based on this, the optimized value evaluation function *L* for the model can be expressed as follows:(3)$$\begin{array}{cc} \; & \min_{d,w,\cdots }\;\;L\left(d,w,\cdots \right)=E_{valid}\left(d,w,\cdots \right)\times {\left[\frac{C(d,w,\cdots )}{C_{0}}\right]}^{\xi } \end{array}$$
where $E_{valid}$ represents the validation loss, C represents the complexity evaluation function, which calculates the model’s FLOPs, $C_{0}$ indicates the FLOPs of the starting point model, and ξ is an exponential weight factor. Given the substantial reduction in search space, we propose a gradient-assist grid search (GAGS) algorithm for multi-objects optimization, as presented in Algorithm 1.

In this algorithm, the starting point of the search can be determined empirically or from a small-scale initial search space. Once the model value evaluation function of all adjacent grid points at the search starting point is determined, the partial derivative at the starting point can be calculated, and the search will proceed along the direction of the partial derivative until a local minimum point is reached. The grid can be further refined based on practical requirements.
**Algorithm 1:** Gradient-assist grid search
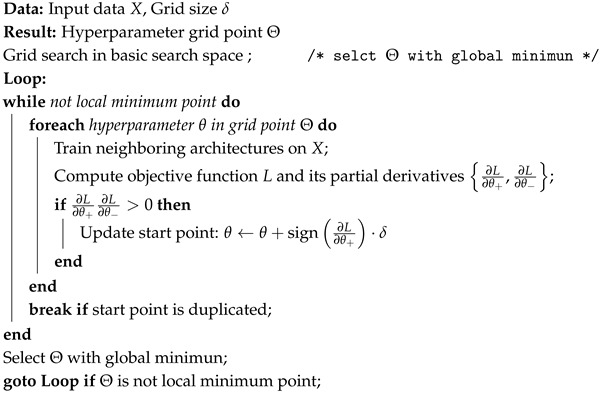


## 3. Simulation Modeling

### 3.1. Sensor Noise

Spot patterns obtained from the image sensor are contaminated by noise. To enhance the robustness of DLWFS, sensor noise is introduced into the simulated data. The total noise of the image sensor encompasses dark noise, photon noise, and readout noise, etc. When the optical intensity is weak, signal-independent noise (such as readout noise) takes domination with a small and constant noise standard deviation $\sigma_{c}$. Conversely, at high optical intensities, signal-dependent noise (such as photon noise and dark noise) becomes predominant, exhibiting a proportionality to the γ power of the input pattern. Thus the total noise η of a typical image sensor pixel can be defined as a piecewise function of the pattern *I* and Gaussian noise N with zero mean and unit variance, modeled as [22](4)$$\eta_{ij}=\left\{\begin{array}{cc} \sigma_{c}N_{ij}, & I_{ij}\leq B\\ \sigma_{u}I_{ij}^{\gamma }N_{ij}, & I_{ij}>B \end{array}\right.$$
where *B* represents the threshold value of the normalized light intensity within the sensor’s dynamic range (approximately $e^{-5}$), which depends on the image sensor. The incident light intensity distribution undergoes photoelectric conversion to obtain the spot image. During the linear phase of photoelectric conversion, the noise distribution can be considered consistent with this distribution. It is evident that in the stage where photoelectric conversion approaches saturation, the noise is suppressed.

### 3.2. Wavefront Distortion

In our simulation, a two-dimensional phase screen is utilized to emulate the wavefront aberration caused by turbulence along the propagation path. To ensure the inclusion of infinite-order Zernike modes, we employ the power spectrum inversion method [23], which entails random sampling of the power spectral density in the complex domain, subsequently followed by a Fourier inverse transformation to generate the phase screen. Typically, additional sampling of low-frequency components is performed to supplement components with periods exceeding the screen size. For the modified von Karman model, the power spectral density of vertical phase fluctuations is given by [24](5)$$\Phi_{w}\left(\kappa \right)=0.023r_{0}^{-5/3}\frac{\exp(-{\left(\kappa /\kappa_{m}\right)}^{2})}{{\left(\kappa^{2}+\kappa_{0}^{2}\right)}^{11/6}},\;\kappa_{0}∼\frac{1}{L_{0}},\kappa_{m}=\frac{5.92}{l_{0}}$$
where $r_{0}$ is the Fried parameter. In the plane wave approximation we have (6)$$r_{0}={\left[0.42k^{2}\sec\zeta \int_{0}^{L}C_{n}^{2}\left(z\right)dz\right]}^{-3/5}$$
where ζ is the zenith angle of optical link, $C_{n}^{2}\left(z\right)$ represents the distribution of the refractive index structure constant along the propagation path *z*, *k* is the wave vector, and *L* is the path length. $L_{0}$ is the outer scale of turbulence, and $l_{0}$ is the inner scale of turbulence.

### 3.3. Simulation Setting

The simulated dataset is used to train and evaluate the CNN model. Turbulence induced wavefront distortion are simulated by the method described in Section 3.2. Because the receiving aperture *D* of the satellite and ground terminals is generally about 0.5 m [25,26], this value is used in the simulations. The ratio $D/r_{0}$ is used to characterize turbulence strength, and nine representative values are selected to describe weak to strong turbulence conditions: 1, 2, 3, 4, 5, 6, 8, 10, 15. In this context, the value of $r_{0}$ is chosen relatively uniformly from 3 cm to 50 cm. The corresponding Zernike coefficients of the wavefront are obtained by least squares (LS) fitting. The spot image reflects the intensity distribution of the optical field after propagating through the receiving lens, with a defocus factor set at β=6. The size of phase screen and spot image are set to 128 × 128 and 64 × 64, respectively. The total sensor noise is added according to the noise model described in Section 3.1 ($\sigma_{c}=0.002$, $\sigma_{u}=0.02$, γ=0.5). Finally, to obtain the input of CNN, spot data are normalized to [0, 1]. Python libraries NumPy and Aotools [27] are used for scientific calculations and turbulence simulations.The total number of datasets is 135,000, with an average signal-to-noise ratio (SNR) of 25.6 dB, which is divided into training, validation, and test sets in proportions of 8:1:1.

## 4. Validation of the Method

### 4.1. Model Optimization: EfficientNet Prototype

We select EfficientNet-B0 [18] as a starting point for our design. The whole WFSNet is structured into three distinct phases: channel expansion, depthwise convolution, and channel compression. Figure 2 presents a schematic representation of the mobile inverted bottleneck convolution (MBConv) block [28] within the model. Compared to traditional convolutions, the dense connections between channels are decomposed into pointwise convolutions and single-channel spatial convolutions, which significantly enhances connectivity efficiency. Additionally, the squeeze-and-excitation (SE) layer [29] incorporates a channel attention mechanism at an exceptionally low computational expense. During the squeeze phase, channel features are extracted by global average pooling, while output channel weights are determined through excitation. Under this circumstance, an additional hyperparameter depthwise convolution expansion ratio *e* is introduced, alongside depth *d* and width scaling ratio *w*.

Next, we determine the settings for the search grid based on our specific requirements. Note that EfficientNet-B0 is suitable for processing 224 × 224 image inputs, while empirically the range of CCD spot spread is often narrower. For an input image size of 64 × 64, after scaling the model as suggested in the literature, the starting grid point for the search is set to be h(3,2.35,3), which results in a small model containing only three MBConv blocks. This scaled model is referred to as Base-WFSNet. It is crucial to emphasize that the scaling ratios proposed by the authors may not always represent the optimal selection, given that the optimization of hyperparameters is contingent upon the specific dataset. Therefore, we will further search for better model architectures.

For a single step of searching, it is preferable for the change in structural hyperparameters to be reasonable. Considering our prototype after scaling, the minimum number of channels is 16/2.35≈7. Therefore, setting the interval for w at 0.35 (which is 1/7 of 2.35) generally meets the requirements. The intervals for *d* and *e* can be set to integers of 1 based on their definitions. Thus, we have established the hyperparameter search intervals δ=(1,0.35,1). To expand the search range, a basic search space is chosen before the search begins: (*d*: 3, 4); (*w*: 2.7, 2.35, 2); (*e*: 2, 3, 4). ξ is set to −0.7.

During the training procedure, we selected the Huber function as loss function defined as follows:(7)$$Huber\left(x,\epsilon \right)=\left\{\begin{array}{cc} 0.5\cdot x^{2}, & |x|\leq \epsilon \\ 0.5\cdot x^{2}+\epsilon \cdot \left(|x|-\epsilon \right), & |x|>\epsilon \end{array}\right.$$
where *x* denotes the error between the model output vector and the true Zernike coefficient vector. It can be seen that the Huber function is actually a combination of mean square error (MSE) and mean absolute error (MAE). Note that MSE loss function expedites the training speed, while the MAE loss function ensures stability in the presence of outliers. The Huber function tends toward MSE for minor errors and MAE for major errors, thereby combining the benefits of both. The constant *h* determines the range designated as outliers, which is set to 1 during our training procedure.

The introduction of severe noises exacerbates the instability of convergence. Following a series of trials, we opted for the Adam optimizer and set the batch size to a relatively large number as 80. Additionally, we introduced a learning rate schedule that decreases linearly from 0.001 to 0.00001 over a span of 300 epochs. The aforementioned measures effectively increase the stability and accuracy of weight updates. All networks were constructed using the TensorFlow 2.9 framework in Python 3.9 and trained on a personal computer equipped with an AMD Ryzen 7 5800H CPU, NVIDIA GeForce RTX 3060 GPU, and 16 GB RAM. TensorFlow’s Lite mobile library is utilized for inference acceleration.

Next, with the GAGS algorithm, the search begin in the initial grid space and will further descend to the stationary point under the guidance of the gradient direction. The start point, optimal point, and six partial derivatives computed throughout the entire search process are detailed in Table 1. The search terminates at h(6,2,4), which has global minimum L value among the evaluated WFSNets, with confirmaton as the local minimum point. Figure 3a presents the L-function values of all evaluated architectures. To visualize the performance, a two-dimensional comparison of validation losses and FLOPs is depicted in Figure 3b. Remarkably, among the architectures with minimal validation loss, h(6,2,4) exhibits the lowest FLOPs. This observation strongly supports the rationality of the objective function L. Consequently, h(6,2,4) is designated as Efficient-WFSNet to highlight its balance between accuracy and speed. Its detailed structural information is illustrated in Figure 4. Two samples demonstrating the model inference results are shown in Figure 5.

The performance of Efficient-WFSNet, Base-WFSNet, and adapted EfficientNet-B0 is verified using the preserved test set, as shown in Table 2. In this context, the training loss, validation loss, and test loss refer to the Huber loss of the model’s inferred first 35 Zernike mode coefficients compared to the true values from the simulated training set, validation set, and test set. We present these losses to evaluate the model’s regression capability in mapping the captured light spots to the 35 Zernike modes of the wavefront at the pupil plane. Since the loss function only captures the errors of the first 35 Zernike coefficients, the root mean square (RMS) error of wavefront is provided to indicate total wavefront sensing errors. Based on the test outcomes, our proposed Efficient-WFSNet shows a 22% improvement in turbulence-induced wavefront sensing accuracy over 35 Zernike modes compared to the Adapted EfficientNet-B0, while it reduces FLOPs and inference time by approximately 80% and 95%, respectively.

### 4.2. Robustness of the Method

In this subsection, we investigate the robustness of real-time DLWFS from two perspectives: turbulence intensity and sensor noise intensity. Figure 6a illustrates the performance variations of Efficient-WFSNet, Base-WFSNet, and adapted EfficientNet-B0 under different turbulence intensities. The red dotted and dashed line indicates the average fitting error of the 35 Zernike modes. Since our model outputs only the Zernike coefficients, using the average fitting error as a benchmark is more meaningful for a reasonable assessment of model performance. As turbulence intensity increases, the model’s inference error gradually rises but does not significantly deviate from the benchmark. At strong turbulence conditions ($D/r_{0}=15$), the average RMS wavefront sensing error of Efficient-WFSNet reaches 0.882 rad (approximately 0.14 λ), which is slightly higher than the benchmark of 0.119 rad (approximately 0.019 λ). 

Figure 6b presents the performance of the three models as a function of sensor noise intensity (from the model in Equation (Equation 4)). Here, we maintained γ=0.5 and scaled $\sigma_{u}$ and $\sigma_{c}$ by the same proportion to achieve a SNR variation from 15 dB to 40 dB. It is evident that when the SNR exceeds 20 dB, the model’s inference error remains stable. At this stage, the average RMS wavefront sensing error of Efficient-WFSNet over $D/r_{0}$ = 1 to 15 remains below the Marechal criterion (1/14λ). This indicates that, for moderate or worse image quality, our method can achieve stable wavefront sensing. However, in the presence of severe noise, the sensing performance deteriorates rapidly due to DLWFS’s reliance on the detailed information of the spot patterns. At SNR = 13.9 dB, the average RMS error of Efficient-WFSNet reaches 0.551 rad (approximately 0.88 λ). Furthermore, it can be observed that when the SNR falls below 15 dB, the performance of Efficient-WFSNet becomes slightly inferior to that of adapted EfficientNet-B0. We attribute this to the fact that the parameter scale of Efficient-WFSNet is significantly smaller than that of adapted EfficientNet-B0 (approximately 1/5), resulting in relatively more unstable connections among hidden features that are more susceptible to noise. 

### 4.3. Experiment Validation

To further validate the feasibility of the wavefront sensing scheme, we conducted an indoor experiment. The setup is depicted in Figure 7. The 638 nm monochromatic laser beam was generated by a single-mode fiber-coupled source (Src, SFOLT FC-638-100-SM), and emitted to free space by a fiber collimator (FC). To mitigate scattering and aberrations, the beam was confined by an iris diaphragm (D). Following lightpath adjustments with a reflective mirror (M), the beam was modulate by a spatial light modulator (SLM, HOLOEYE LC-2500). A polarizer (P) was employed to adjust the polarization direction to the optimal angle for the liquid crystal on silicon (LCoS). Lens L1 and L2 expanded the beam to encompass the LCoS surface. Phase screens were projected onto the LCoS, and the distorted wavefront was received by objective lens L3. A commercial charge coupled device (CCD) was positioned behind the lens to capture the defocused spot.

When applying a data-based wavefront sensing scheme to a new setup, calibration is imperative. We generated a dataset containing of 2000 digital phase screens and executed transfer learning on pre-training models with spot images captured from the CCD. The majority of the model parameters are frozen, with only the fully connected layers in the head block undergoing optimization. Consequently, this transfer learning process can be succinctly described as follows:(8)$$\min_{\omega ,b}\left|\right|f_{M}\omega +b-a_{N}{\left|\right|}^{2}$$
where $f_{M}$ denotes n samples of M-dimensional features, $a_{N}$ signifies n samples of N order Zernike coefficients, ω represents an M × N matrix, and *b* corresponds to an N-dimensional offset. Here, M depends on the model architecture while N remains a constant at 35. This optimization problem can be efficiently resolved by LS method, requiring only a brief period of seconds on our hardware platform.

An indoor wavefront sensing experiment is conducted under moderate turbulence simulated by a SLM. Here, we utilized an infinite phase screen [30] to represent wavefront distortion. The pseudo wind speed $v_{w}$ is set at 15 m/s, and $D/r_{0}=2$. Since the modulation speed of the SLM is merely 60 Hz, for the adapted EfficientNet-B0 model, an inference time of around 5 ms introduced a one-frame lag. The experiment results is demonstrated in Figure 8, with detailed information listed in the Table 3.

### 4.4. Disscusion

Generally, our methodology substantially improves existing CNN prototypes for real-time DLWFS. However, being a data-driven method, it does come with certain limitations. Optical system aberrations, defocus deviations, camera exposure time, and pixel size can introduce input distribution deviations, thereby impacting the performance of the pre-trained CNN. As a result, additional calibration training is required. We propose a simplified calibration approach: since aberrations and defocus deviations can be represented by a fixed bias in Zernike coefficients, we freeze the parameters of the convolutional layers and optimize only the linear transformation from features to outputs (i.e., the fully connected layer in the head block). Our experiments have confirmed the effectiveness of this approach, although overcoming this drawback entirely remains challenging.

Additionally, image quality is critical for sensing performance in image-based wavefront sensing methods. When the SNR of the light spot is low, intense noise distorts the intensity distribution, leading the CNN to capture incorrect information. To mitigate this issue, we introduced significant noise (with an SNR of approximately 25 dB) during model training, allowing the model to adapt to distortions caused by image sensor noise. In Section 4.2, we provide a demonstration, through a series of simulations at SNRs ranging from 15 to 40 dB, that our sensing scheme maintains stable performance when image quality is moderate or worse (greater than 20 dB). However, as image quality continues to deteriorate, sensing accuracy will inevitably decline rapidly.

## 5. Conclusions

In this paper, a real-time DLWFS method for dynamic stochastic wavefront distortion was proposed, which demonstrates the significant potential for improvement of the CNN prototype in DLWFS. We present an effective automated model structure design and optimization algorithm, GAGS, which led to the development of a compact network, Efficient-WFSNet, based on the EfficientNet-B0 prototype. Our method significantly overcomes the inherent problem of the great computational demand of deep CNNs while improving accuracy. To our knowledge, as a CNN-based method, Efficient-WFSNet achieves an unprecedented inference time of only 0.2 ms, which greatly reduces servo lag and thus reduces measurement error. Both simulated and experimental results show the superior performance of Efficient-WFSNet, an outcome primarily attributed to its network architecture, which is suitable for turbulence-induced wavefront sensing tasks. This method significantly overcomes the high computational demands associated with deep CNNs while enhancing the accuracy of wavefront reconstruction. Our research provides valuable insights for wide applications of real-time DLWFS in SGLC systems, offering advantages such as low cost, high precision, and low power consumption. It represents an intelligent, algorithm-based approach to real-time adaptive wavefront compensation, opening new avenues for optical wavefront correction, which holds significant importance.

## Figures and Tables

**Figure 1 sensors-25-00480-f001:**
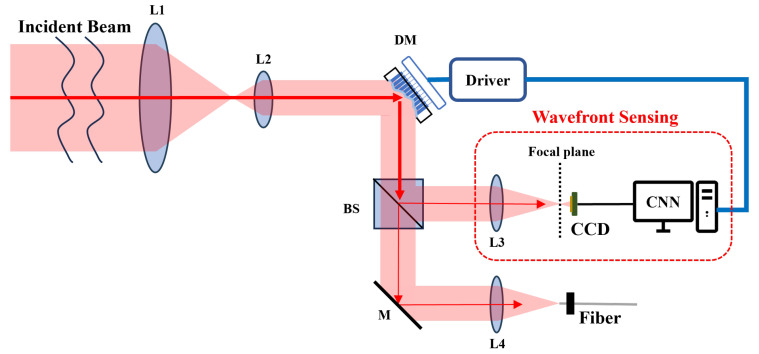
Schematic diagram of the DLWFS scheme for AO correction in communication subsystem.

**Figure 2 sensors-25-00480-f002:**
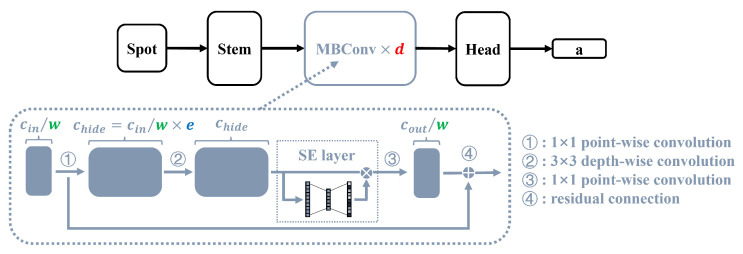
Schematic diagram of WFSNet and MBConv block.

**Figure 3 sensors-25-00480-f003:**
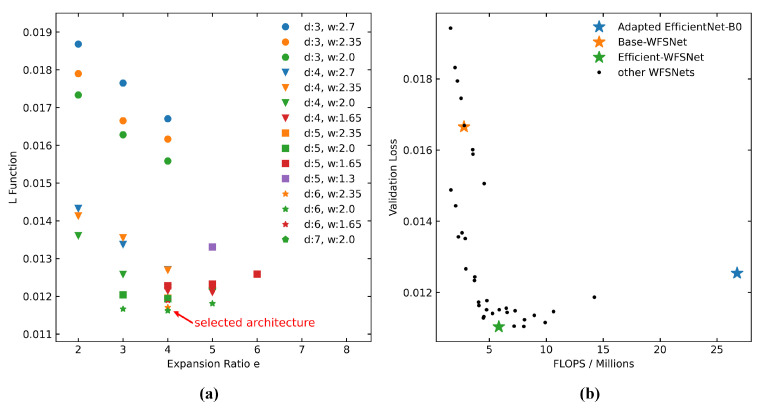
Comparison of network architectures performance. (**a**) Objective function L (Equation (Equation 3)) of WFSNets, where marker style indicates *d*, color indicates *w*, and *x* coordinate indicates *e*. The selected architecture is h(6,2,4). (**b**) FLOPs and validation loss comparison among WFSNets, in addition to Adapted EfficientNet-B0.

**Figure 4 sensors-25-00480-f004:**
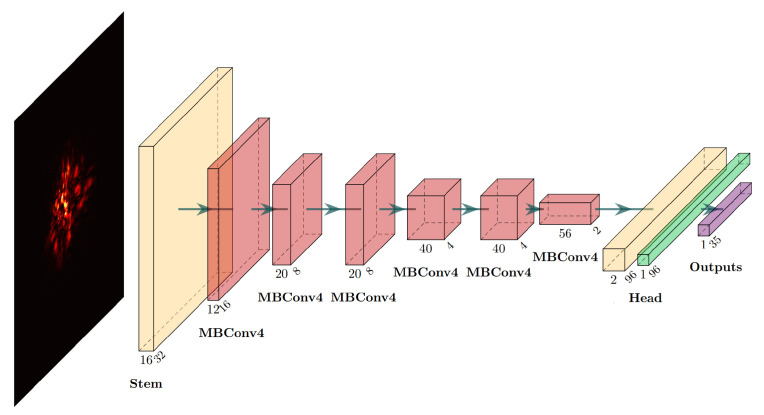
Structural diagram of Efficient-WFSNet. The architecture of MBConv is detailed in Figure 2, where the notation ‘4’ signifies that the 1 × 1 point convolution expands the number of channels by a factor of 4. All convolutional layers are followed by batch normalization, and the activation function utilized in both convolutional and fully connected layers is the Swish function.

**Figure 5 sensors-25-00480-f005:**
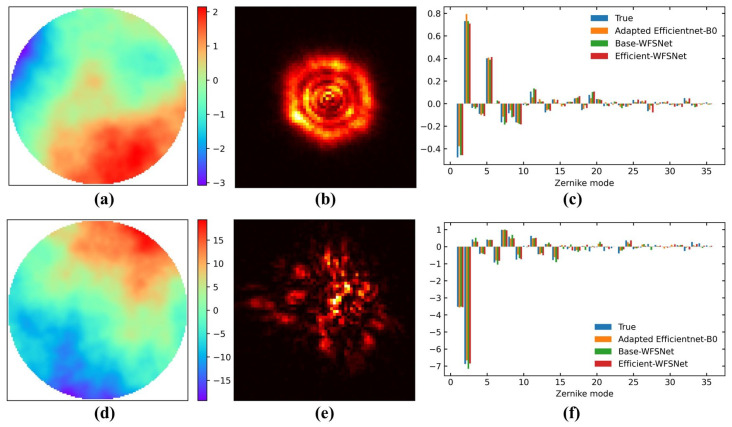
Two samples demonstrating the 35 Zernike mode inference performance of the prototype, Base-WFSNet, and Efficient-WFSNet. Phasescreens (**a**,**d**), normalized spots (**b**,**e**), and performance comparisons (**c**,**f**) are presented. Top row: $D/r_{0}=1$. Low row: $D/r_{0}=15$.

**Figure 6 sensors-25-00480-f006:**
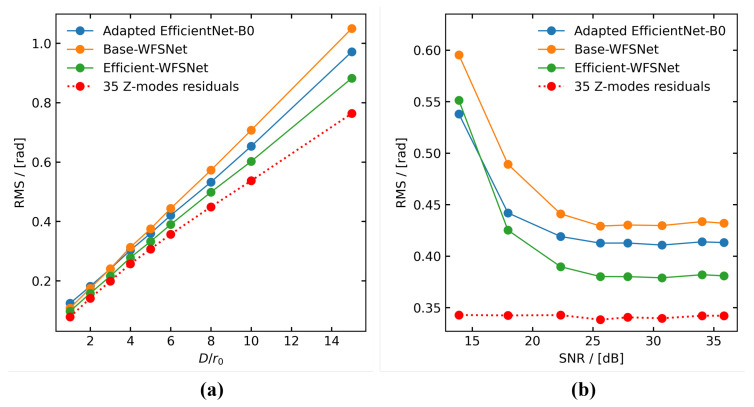
The performance of the three models under different (**a**) turbulence intensity and (**b**) sensor noise intensity. The red dotted and dashed line indicates the average fitting error of the 35 Zernike modes. The seven pairs of values $\left(\sigma_{u},\sigma_{c}\right)$ in (**b**) are as follows: (0.002, 0.0002), (0.005, 0.0005), (0.01, 0.001), (0.015, 0.0015), (0.02, 0.002), (0.05, 0.005), (0.08, 0.008). The corresponding SNR ranges from 35.9 dB to 13.9 dB.

**Figure 7 sensors-25-00480-f007:**
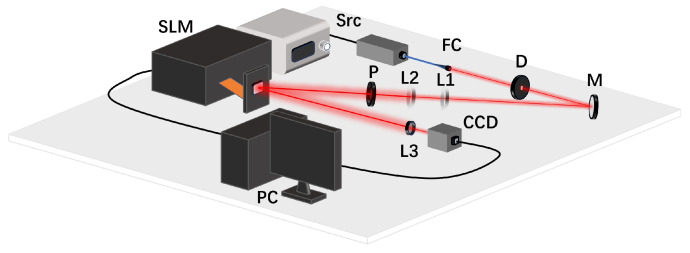
Experimental setup diagram: Src, 638 nm laser source; FC, fiber collimator; D, iris diaphragm; M, reflective mirror; L1, lens (diameter 20 mm, focal length 10 mm); L2, lens (diameter 20 mm, focal length 50 mm); L3, objective lens (diameter 5 mm); P, polarizer; SLM, spatial light modulator; CCD, charge coupled device; PC, personal computer.

**Figure 8 sensors-25-00480-f008:**
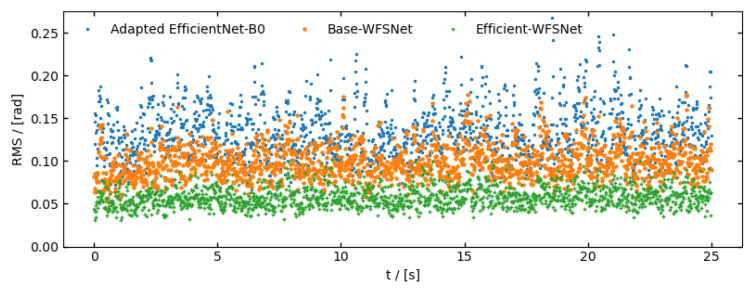
RMS loss of 35 Zernike modes in continuous wavefront sensing experiment under simulated moderate turbulence.

**Table 1 sensors-25-00480-t001:** Partial derivatives of grid neighborhood in the search process.

Step	0	1	2	3	4
Start point	(3,2.35,3)	(4,2,4)	(5,1.65,5)	(5,2,4)	(6,2,4)
$\partial L/\partial d_{+}$ ($\times 10^{-4}$)	\	−2.91	−1.32	3.49	3.33
$\partial L/\partial d_{-}$ ($\times 10^{-4}$)	\	−33.6	2.17	−2.91	−3.10
$\partial L/\partial w_{+}$ ($\times 10^{-4}$)	\	4.68	−0.620	0.157	0.869
$\partial L/\partial w_{-}$ ($\times 10^{-4}$)	\	0.763	−9.82	−3.49	−2.74
$\partial L/\partial e_{+}$ ($\times 10^{-4}$)	\	−0.905	2.67	0.328	1.87
$\partial L/\partial e_{-}$ ($\times 10^{-4}$)	\	−3.56	0.410	−1.09	−0.435
Adjacent best point	(4,2,4)	(5,2,4)	(4,1.65,5)	(6,2,4)	(6,2,4)
Adjacent best L ($\times 10^{-2}$)	1.223	1.193	1.211	1.162	1.162

**Table 2 sensors-25-00480-t002:** Comparison of model performance, parameters, and inference speed.

	Params	Flops	Inference Time (ms)	Training Loss (rad)	Validation Loss (rad)	Test Loss (rad)	Test RMS (rad)
Adapted EfficientNet-B0	324,644	26.75 m	5.315	0.2261	0.2278	0.2307	0.4207
Base-WFSNet	13,869	2.77 m	0.122	0.2681	0.2713	0.2733	0.4430
Efficient-WFSNet	67,766	5.83 m	0.204	0.1756	0.1774	0.1793	0.3906

**Table 3 sensors-25-00480-t003:** Comparison of model performance with experiment data.

	Training Loss (rad)	Test Loss (rad)	Test RMS (rad)
Adapted EfficientNet-B0	0.0510	0.1325	0.2179
Base-WFSNet	0.0651	0.0991	0.2032
Efficient-WFSNet	0.0399	0.0580	0.1918

## Data Availability

The data presented in this study are available on request from the corresponding author.

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
