# Peer review of "Enhanced Neural Architecture for Real-Time Deep Learning Wavefront Sensing"

_sensors, 2025, doi:10.3390/s25020480_

Round 1

Reviewer 1 Report

Comments and Suggestions for Authors

In the paper, the authors use Deep-learning based wavefront sensing method for applications in Satellites optical communications through the atmosphere. The authors propose to use convolutional neural networks with a novel Multi-Objective Architecture search scheme aiming at exploring and selecting the best parametrization of the method, by minimizing a merit function. The paper mentions the performance of this new scheme is improved by 80% in computational time and 22% in accuracy with respect to a reference standard Deep-learning scheme.

Globally the paper is interesting and deserves publication pending some modifications.

Comments

1.      Line 16: it is mentioned SLC have resistance to interference. This formulation is misleading as optical link do have interference. The difference with RF lies in the fact that optical links have much stronger beam directivity with respect to RF links, therefore no interference occur. The sentence should be modified.

2.      P2/10, in the introduction, the structure of the paper is mentioned e.g “In Chapter 1, a real-time ….”. But it is not clear what is meant by “Chapter”. There are no chapters in the paper. So it is assumed “Chapters” correspond to sections but it is also not the case as there are no as many Chapters as sections and they also don’t seem to have the same name. This should be modified.

3.      Line 59, it is mentioned “For the first time, we achieved….”. it should be clarified whether it is a first world-wide or whether it is a first for the authors.

4.      The paper mentions the benefits of the methods. But there are also drawbacks, which should be clearly highlighted in the same places as where the benefits are mentioned.

5.      Line 77: it is mentioned the light is focused on a DM. But this is not the case. The DM is in a pupil plane, not in a focal plane. Please correct.

6.      Figure 1 shows a feedforward configuration. This is not a typical configuration for an Adaptive Optics system. The use of this configuration should therefore be 1/ mentioned and 2/ properly justified.

7.      The definition of the quantities used should be improved. At least: all the figures of merit should be properly defined and explained, e.g the paper mentions train loss, Validation loss, Test loss, Test RMS, Experimental loss, Experiment RMS. These quantities should be clearly defined, otherwise it makes impossible to quantify the benefits of the proposed method and makes the reading cumbersome.

8.      Line 176: it is mentioned the definition of r0 is for plane wave but it is not really true. The provided definition is for horizontal links with constant Cn2. This should be mentioned and better explained.

9.      Results about the robustness of the method should be provided. At least one a discussion in the conclusion should be added.

10.  Figure 4: The caption of the figure could be improved. It would be clearer to say “top raw: D/r0 = 1. Low row: D/r0 = 15” instead of repeating it 3 times.

11. In the experimental results sections, it seems the inference time is not calculated, while it is the main argument of the method. Therefore, it should be clarified everywhere (Introduction, core of the text and conclusion) that the inference time is only evaluated numerically and not experimentally. Or the experimental result of the inference time should also be provided.

Author Response

Comments 1: Line 16: it is mentioned SLC have resistance to interference. This formulation is misleading as optical link do have interference. The difference with RF lies in the fact that optical links have much stronger beam directivity with respect to RF links, therefore no interference occur. The sentence should be modified.

Response 1: Thank you for pointing out our errors; we fully agree with your perspective. Compared to radio frequency communication, satellite laser communication exhibits superior directivity, providing resistance to electromagnetic interference and improved confidentiality. However, it does not demonstrate particularly significant advantages in resisting atmospheric turbulence interference. The expression you suggested regarding good directivity is more appropriate than simply stating resistance to interference. Consequently, this statement has been revised in Page 1, Line 16 to: "better beam directivity.

Comments 2: P2/10, in the introduction, the structure of the paper is mentioned e.g “In Chapter 1, a real-time ….”. But it is not clear what is meant by “Chapter”. There are no chapters in the paper. So it is assumed “Chapters” correspond to sections but it is also not the case as there are no as many Chapters as sections and they also don’t seem to have the same name. This should be modified.

Response 2: We are very sorry for our incorrect writing. We completely appreciate your suggestion and have removed the misleading term "Chapter" from the Introduction. Additionally, we have adjusted the phrasing accordingly. The revised content can be found on Page 2, Line 47.

Comments 3: Line 59, it is mentioned “For the first time, we achieved….”. it should be clarified whether it is a first world-wide or whether it is a first for the authors.

Response 3: Thank you for pointing this out. We apologize for the lack of rigor in our expressions and the confusion it may have caused. The statement in Line 59 is a reflection of our current understanding of world-wide research. Among the recently referenced deep learning-based wavefront sensing approaches, none—to our knowledge—has achieved an inference time below 1 ms. Specifically, the methods cited include Zhou et al. [10]'s phase diversity technique (370 ms), Li et al. [11]'s untrained approach (80 s), Paine et al. [12]'s work (200 ms), Nishizaki et al. [13]'s method (9.2 ms), Guo et al. [14]'s approach (12 ms), Wang et al. [15]'s method (8.7 ms for adapted EfficientNet-B0), and You et al. [16]'s method (8 ms for adapted Xception). In contrast, our system achieves an inference time of 0.12 ms with the Efficient-WFSNet (Table 3), and it operates on a computational platform (AMD Ryzen 7 5800H CPU, NVIDIA GeForce RTX 3060 GPU, and 16 GB RAM) that is not markedly more advanced than those works. We hope this response is convincing, and we have revised the phrasing to enhance clarity and eliminate ambiguity, which can be found on Page 2, Line 59.

Comments 4: The paper mentions the benefits of the methods. But there are also drawbacks, which should be clearly highlighted in the same places as where the benefits are mentioned.

Response 4: We greatly value the issue you raised and have supplemented our explanations and discussions based on your suggestions. Overall, our approach significantly optimizes existing CNN prototypes for real-time DLWFS. However, as a data-driven method, it has certain limitations:

  1. Aberrations in the optical system, deviations in defocus, exposure time of camera, and the size of the camera pixels can cause deviations of the input distribution, thereby affecting the performance of the pre-trained CNN. Consequently, extra calibration training is necessary. We have designed a simplified calibration scheme: since the aberrations and defocus deviations can be represented by a fixed bias of Zernike coefficients, we freeze the parameters of the convolutional layers and only optimize the linear transformation from features to outputs (i.e., the fully connected layer in the Head block). Our experiments have confirmed the effectiveness of this calibration scheme, but it remains challenging to completely overcome this drawback.
  2. As an image-based wavefront sensing method, image quality is crucial for sensing performance. When the SNR of the light spot is low, severe noise alters the intensity distribution, leading the CNN to capture erroneous information. To address this, we introduced significant noise (with an SNR of approximately 25 dB) during the model training process, enabling the model to adapt to image distortions caused by image sensor noise. In Section 4.3, we provide additional information demonstrating through a series of simulations at SNRs ranging from 15 to 40 dB that when the image quality is moderate or worse (greater than 20 dB), our sensing scheme can maintain stable performance. However, as image quality continues to deteriorate, the sensing accuracy will inevitably decline rapidly.

Related content can be found in "4.4 Discussion" on Page 11.

Comments 5: Line 77: it is mentioned the light is focused on a DM. But this is not the case. The DM is in a pupil plane, not in a focal plane. Please correct.

Response 5: Thank you for pointing out our mistakes. The section has been rewritten and corrected, and you can find it on Page 2, Line 77.

Comments 6: Figure 1 shows a feedforward configuration. This is not a typical configuration for an Adaptive Optics system. The use of this configuration should therefore be 1/ mentioned and 2/ properly justified.

Response 6: Thank you for pointing this out. We have revised Figures 1 and descriptions on Page2, Section 2.1 in accordance with your suggestions.

Comments 7: The definition of the quantities used should be improved. At least: all the figures of merit should be properly defined and explained, e.g the paper mentions train loss, Validation loss, Test loss, Test RMS, Experimental loss, Experiment RMS. These quantities should be clearly defined, otherwise it makes impossible to quantify the benefits of the proposed method and makes the reading cumbersome.

Response 7: Thank you for pointing out the issues in the manuscript. We apologize for the inappropriate naming and the difficulties it has caused in reading. We have added supplementary explanations and corrected the terminology:

  1. We divided the simulated dataset into three independent and identically distributed datasets: training set, validation set, and test set. The training set is used to update the model parameters; the validation set is for hyperparameter tuning and comparing model performance in MANS process; and the testing dataset is used for final evaluation of CNN performance after training. In the experiments, since there were no hyperparameters that needed further adjustment during the calibration process, the dataset was divided into two independent and identically distributed datasets: training set and test set.
  2. The training loss, validation loss, test loss, and experimental loss refer to the Huber loss of the CNN's inferred first 35 Zernike mode coefficients compared to the true values from the simulated training set, validation set, test set, and the experiment training set. We present these losses to evaluate the model's regression capability in mapping the captured light spots to the 35 Zernike modes of the wavefront at the pupil plane. You can find and explanations at Page 7, Line 260.
  3. The test RMS and experimental RMS refer to the RMS error between the fitted wavefront and the true wavefront from the simulated test set and the experiment test set. We present these errors to help readers better understand and evaluate the performance of wavefront sensing. You can find and explanations at Page 7, Line 264.

Additionally, your reminder made us realize that the terms "experimental loss" and "experimental RMS" were not accurate; therefore, we have corrected them to "test loss" and "test RMS" with further explanations. You can find and corrections at Page 11, Table 3.

Comments 8:  Line 176: it is mentioned the definition of r0 is for plane wave but it is not really true. The provided definition is for horizontal links with constant Cn2. This should be mentioned and better explained.

Response 8: Thank you for pointing out the aspects we overlooked. We have revised the expression for r0 in accordance with your suggestions to better align it with our research context. You can find the updated expression in Equation 6 on Page 5.

Comments 9: Results about the robustness of the method should be provided. At least one a discussion in the conclusion should be added.

Response 9: Thank you for raising this question. In response to your concerns, we conducted a simulation analysis of the robustness of our method from two aspects: turbulence intensity and sensing noise intensity.

  1. As turbulence intensity increases, the model's inference error gradually rises but does not significantly deviate from the average fitting RMS error of 35 Zernike modes. Since our model outputs only the Zernike coefficients, we use the average fitting error as a benchmark is more meaningful for a reasonable assessment of model performance. As turbulence intensity increases, the model's inference error gradually rises but does not significantly deviate from the benchmark.
  2. The simulation results also confirm that when the SNR exceeds 20 dB, the model's inference error remains stable. This indicates that our method can achieve stable wavefront sensing for moderate and worse image quality.

We have added a subsection titled "4.2 Robustness of the Method" to discuss this issue in detail, which can be found on Page 9.

Comments 10: Figure 4: The caption of the figure could be improved. It would be clearer to say “top raw: D/r0 = 1. Low row: D/r0 = 15” instead of repeating it 3 times.

Response 10: Thank you very much for your suggestion! We have revised the caption of Figure 5 on Page 9 according to your advice.

Comments 11: In the experimental results sections, it seems the inference time is not calculated, while it is the main argument of the method. Therefore, it should be clarified everywhere (Introduction, core of the text and conclusion) that the inference time is only evaluated numerically and not experimentally. Or the experimental result of the inference time should also be provided.

Response 11: Thank you for raising this concern. We believe that the main factors limiting the sensing latency in DLWFS is the sum of the following two components: the time delay of which the camera acquires and uploads images, and the inference time delay of the CNN. In this paper, we focus on reducing the significant computational burden of the CNN to alleviate the delay caused by the latter, which is believed solely related to the CNN's FLOPs and the computing platform, remaining consistent in both simulations and experiments.

Our camera can achieve a maximum frame rate of 240 frames per second in grayscale mode. Furthermore, common high-speed cameras can easily achieve high frame rates. Such as the Cyclone-1HS-3500 from Optronis can achieve a maximum frame rate of 12,385 fps (0.08 ms) at a resolution of 320 x 240 (while our input size is 64 x 64). Therefore, compared to the model inference time, we believe that the camera's measurement latency will not be the most important limiting factor of sensing speed. The actual sensing speed in the experiment is limited to 60 Hz due to the constraints imposed by the SLM modulation frequency, as already noted on Page 11, line 324.

Finally, we would like to emphasize that take your concerns seriously. Thus, we have revised some statements in the Introduction and Conclusion to enhance their rigor: “the optimized WFSNet achieves inference times significantly below 1 ms (approximately 0.2 ms)” and “Efficient-WFSNet achieves an unprecedented inference time of only 0.2 ms.” You can find these revisions in the re-submitted manuscript.

Reviewer 2 Report

Comments and Suggestions for Authors

The manuscript introduces an enhanced wavefront sensing neural network (WFSNet) based on convolutional neural networks (CNNs). It aims to achieve real-time deep learning wavefront sensing (DLWFS) for dynamic random wavefront distortions caused by atmospheric turbulence. This approach has significant practical value in the field. However, there are several suggestions for improvement:

1. Ensure that full English terms are provided only when an acronym appears for the first time in the text and use the acronym consistently thereafter. Carefully review the manuscript to avoid repeated use of full terms such as "CNN" and "SGLC" in the abstract and introduction.

2. Provide a detailed explanation of how the threshold value B in Section 3.1 is determined.

3. Clarify how the nine representative values (1, 2, 3, 4, 5, 6, 8, 10, 15) in Section 3.3 are specifically defined to represent the range of moderate to strong turbulence.

4. Explain the rationale for choosing the Adam optimizer.

5. Convert Table 2 in Section 4.1 into a figure to more clearly illustrate the relationship between different convolutional layers.

6. Specify the activation function used in the neural network.

Author Response

Comments 1:  Ensure that full English terms are provided only when an acronym appears for the first time in the text and use the acronym consistently thereafter. Carefully review the manuscript to avoid repeated use of full terms such as "CNN" and "SGLC" in the abstract and introduction.

Response 1 Thank you for pointing out the errors in the manuscript. The repeated acronym "CNN" in Line 43 on Page 2, “WFSNet” in Line 201 on Page 6 has been removed. To avoid any misunderstandings, our understanding is that if an acronym has already been introduced in the abstract, it typically needs to be redefined in the main text as well, for instance, https://doi.org/10.3390/s25010264 (randomly picked). Thank you once again for bringing this to our attention. We hope that we have corrected all the errors.

Comments 2:  Provide a detailed explanation of how the threshold value B in Section 3.1 is determined.

Response 2: Thank you for your question. B is the threshold value of the normalized light intensity within the dynamic range of the sensor: below this threshold, the internal noise is dominant; above this threshold, the signal-dependent noise is dominant. The value of B depends on the type of image sensor, which is determined by practical calibration. This part is referenced from [22] https://doi.org/10.1109/TIP.2006.877363, and more relevant descriptions is in “â…¡.A. CCD Noise Model”.

We rewrite this part as follows: “where B is the critical value of the normalized light intensity in the dynamic range of the sensor (approximately e-5), which depends on the image sensor”, Section 3.1, Page 5 of the manuscript

Comments 3: Clarify how the nine representative values (1, 2, 3, 4, 5, 6, 8, 10, 15) in Section 3.3 are specifically defined to represent the range of moderate to strong turbulence.

Response 3: Thank you for raising this valuable question. This set of values was selected because we chose r0 relatively uniformly (ranging from 50 cm to 3 cm, which reflects turbulence conditions from weak to strong), resulting in an uneven distribution for D/r0 (with integer values selected as much as possible). Here, we assume that D is 0.5 m, for instance, in ESA's OPALE terminal (aperture of 25 cm) and OGS ground terminal (receiving aperture of 1 m) [25], as well as in NASA's LCRD project (satellite terminal aperture of 10.8 cm and ground terminal receiving aperture of 60 cm) [26]. We have added relevant references, which can be found in the references section of the manuscript.

To clarify this point, we have corrected and rewritten the relevant content in Section 3.3 on Page 5.

Comments 4: Explain the rationale for choosing the Adam optimizer.

Response 4: We appreciate your insightful question regarding our choice of the Adam optimizer. The reasons for our selection are as follows:

1. During the initial phase, we also experimented with other optimizers, such as SGD and fixed learning rate strategies, testing them on both the Adapted EfficientNet-B0 and Base-WFSNet models. However, the results indicated that the loss was difficult to decrease consistently and exhibited severe oscillations. Consequently, we continued to adjust our optimization methods and ultimately found that the combination of the Adam optimizer with a learning rate decay strategy yielded the best results. We think that these attempts were somewhat trivial and beyond the scope of this study, which is why they are not detailed in the manuscript.

2. The Adam optimizer incorporates both first-order and second-order momentum corrections of gradient, significantly enhancing the optimization efficiency of deep models. As noted, we require extensive training for WFSNet model ls and need to optimize structural hyperparameters based on their performance. The use of the Adam optimizer has greatly improved the iteration efficiency of our multi-objective neural architecture search process.

Comments 5: Convert Table 2 in Section 4.1 into a figure to more clearly illustrate the relationship between different convolutional layers.

Response 5: Thank you very much for your suggestion. We initially believed that Figure 2 in Section 4.1 adequately illustrated the fundamental structure of WFSNet, which is why we only used Table 2 to present the architecture of Efficient-WFSNet. Your feedback has helped us realize that the distance between them made it difficult to establish a connection. Therefore, following your recommendation, we have replaced Table 2 in Section 4.1 with Figure 4 on Page 8.

Comments 6: Specify the activation function used in the neural network.

Response 6: Thank you for pointing this out. The activation function used in the Stem, all MBConv blocks, and the Head, is Swish, defined as Swish(x) = x∙sigmoid(x). Compared to the ReLU activation function, Swish is a smooth function with a continuous derivative. Moreover, its derivative exists when x < 0 , which helps prevent neuron failure. This characteristic enables the gradient descent algorithm to achieve better convergence during the optimization process.

You can find the additional information in the caption of Figure 4 on Page 8.
